# The Effects of a Patient-Specific Integrated Education Program on Pain, Perioperative Anxiety, and Functional Recovery following Total Knee Replacement

**DOI:** 10.3390/jpm12050719

**Published:** 2022-04-29

**Authors:** Cheng-Jung Ho, Yen-Ti Chen, Hung-Lan Wu, Hsuan-Ti Huang, Sung-Yen Lin

**Affiliations:** 1Department of Orthopaedic Surgery, Kaohsiung Medical University Hospital, Kaohsiung Medical University, Kaohsiung 80756, Taiwan; rick_free@mail2000.com.tw (C.-J.H.); hthuang@kmu.edu.tw (H.-T.H.); 2Departments of Orthopaedics, School of Medicine, College of Medicine, Kaohsiung Medical University, Kaohsiung 80756, Taiwan; 3Orthopaedic Research Center, Kaohsiung Medical University, Kaohsiung 80756, Taiwan; 4Regeneration Medicine and Cell Therapy Research Center, Kaohsiung Medical University, Kaohsiung 80756, Taiwan; 5Graduate Institute of Clinical Medicine, Kaohsiung Medical University, Kaohsiung 80756, Taiwan; 6Department of Orthopedics, Kaohsiung Municipal Ta-Tung Hospital, Kaohsiung 80145, Taiwan; e0712712@yahoo.com.tw; 7Nursing Department, Kaohsiung Municipal Ta-Tung Hospital, Kaohsiung 80145, Taiwan; 8Graduate Institute of Long-Term Care, Tzu Chi University of Science and Technology, Hualien 97005, Taiwan; hunglan@ems.tcust.edu.tw

**Keywords:** total knee arthroplasty, integrated education, prehabilitation, knee osteoarthritis

## Abstract

The perioperative care of patients undergoing total knee arthroplasty (TKA) affects functional recovery and clinical outcomes. This study aims to introduce a patient-specific integrated education program (IEP) into the TKA clinical pathway and to evaluate patient outcomes between the intervention and control groups. We performed a two-site, two-arm, parallel-prospective controlled trial. The experiment group received an IEP incorporating verbal preoperative education, prehabilitation, multidisciplinary personalized rehabilitation during hospitalization, and supervised self-executed home-based exercise after discharge. The control group received regular TKA clinical care. We monitored the pain intensity, anxiety scores, and functional scores at six time points from the pre-operation interview to 3 months post-operation. The pain score was significantly decreased in the IEP group during hospitalization (*p* < 0.01) and before discharge (*p* < 0.05). The anxiety status was also improved after intervention in terms of state and trait anxiety inventory scores (*p* < 0.001) during hospitalization. The patient-reported (WOMAC) or physician-reported (American Knee Society Score) functional scores (*p* < 0.01 at most of the time points) all improved significantly under hospitalization. We found that the patient-specific IEP combining preoperative education, prehabilitation, the in-hospital group education class, and postoperative care navigation is effective in reducing postoperative pain, decreasing perioperative anxiety, and facilitating functional recovery following TKA.

## 1. Introduction

Total knee arthroplasty (TKA) is a reliable surgical procedure to relieve pain and restore knee joint function. It improves the health-related quality of life for end-stage knee arthritis patients. Prior studies have shown that the number of TKA procedures has increased annually, and the future demand for TKA is projected to grow worldwide [1,2,3]. However, the length of TKA hospital stays has steadily decreased, and outpatient joint arthroplasty has progressively increased in recent years [4,5]. Shorter hospital stays and the growing trend of outpatient surgery have increased the demand for self-care education.

Previous studies report a gap in knowledge expectations in patients scheduled for joint arthroplasty. Johansson Stark et al. conducted translational research in three Nordic countries and found that 77% of individuals had unfulfilled knowledge [6]. In an international surgical trip mission in Guyana, Solano et al. evaluated the effectiveness of preoperative education classes and found that 78% of patients scored less than 40% on a pre-class knowledge questionnaire [7]. Most of the patients undergoing joint replacement surgery are elderly, and their understanding of preoperative education could be influenced by such factors as inadequate time, the use of medical jargon, or education level. Patients also may not remember the content of preoperative education postoperatively [8].

In this study, we used a patient-specific integrated education program (IEP), including a verbal one-on-one preoperative education consultation and rehabilitation training two weeks before the operation, personalized postoperative rehabilitation and a group education class during the hospital stay, and a care navigation program supervising outpatient exercise after discharge to reinforce the patient’s motivation for rehabilitation and facilitate functional recovery. We hypothesized that an IEP may improve clinical outcomes after TKA compared with the usual care model. This study aims to investigate whether an IEP is associated with less clinical pain, reduced anxiety status, and better functional recovery after TKA.

## 2. Materials and Methods

### 2.1. Study Design

This study was a two-site, two-arm, parallel-group controlled trial with a quasi-experimental research design to evaluate the efficacy of an IEP on TKA-associated pain, anxiety, and functional recovery. The two study sites were hospital A (intervention group) and hospital B (control group). This project was approved by our institutional review board (registration number: KMUHIRB-E(I)-20190255). It was also registered in ClinicalTrials.gov (ID: NCT05346822). Both hospitals are in the same medical system and use the same surgical methods, medical care models, and health education manuals for patients undergoing TKA. All participants were recruited from the outpatient clinics of the senior attending physician.

### 2.2. Participants

The inclusion criteria for this study were as follows: (1) Patients over 65 years of age; (2) scheduled to receive unilateral TKA due to advanced osteoarthritis; (3) able to understand the study and express opinions clearly; (4) willingness to participate in this study and provide informed consent. Exclusion conditions for acceptance were diagnosis of inflammatory joint disease, neurosensory system disease (such as stroke or parkinsonism), intellectual impairment, or dementia. Patients who fulfilled the inclusion criteria in the absence of any exclusion criteria were enrolled in the study, and the investigators arranged the hospital for TKA randomly.

### 2.3. Intervention Group (Hospital A)

The same clinical pathway was carried out in both groups. This is a 5-day clinical pathway, including admission day (1st day) for preoperative preparation, operation day (2nd day), rehabilitation days (3rd and 4th day), and discharge day (5th day). Patients in the intervention group (hospital A) received IEP initiated after the patients’ decision and ending 3 months after the surgery. The details of the IEP program are described below and shown in Figure 1.

Preoperatively: The preoperative education and prehabilitation were conducted one-on-one by an experienced nurse two weeks before TKA. Preoperative education encompassed surgical expectations, possible complications, detailed surgical procedures, pain neuroscience, pain management, postoperative care, and rehabilitation exercises. Due to the inadequate literacy of some patients, the in-person educator used diagrams and photos from actual clinical practice and a knee joint model to explain the whole process of treatment, teach what they should do before the operation, demonstrate the postoperative rehabilitation exercises, and answer any remaining questions.

During hospitalization: Besides the personalized rehabilitation programs, all operated patients and their primary caregivers were encouraged to attend group rehabilitation education on postoperative day 2. The group education class was provided by a multidisciplinary team including a physiotherapist, two nurse practitioners, a nurse care manager, and an orthopedic doctor. The content of the group education class encompassed postoperative care, home-based exercise training, pain control regimens, sharing postoperative experiences, showing successful cases, and forming a patient support group.

After discharge: As postoperative care navigation for TKA patients, the nurse care manager made phone calls to patients 1 week after discharge and then biweekly for three months. In the phone calls, navigators monitored the efficacy of the rehabilitation exercises and also answered questions about the patients’ concerns.

### 2.4. Control Group (Hospital B)

The patients in the control group (hospital B) received the same clinical pathway and the usual care model, including a preoperative information consultation and education with a TKA education manual two weeks before surgery. The rehabilitation programs, including continuous passive range of motion, muscle strengthening exercises, and gait training, were carried out routinely from the first day after surgery. The patients were discharged from the hospital on the 5th day and scheduled for follow-up at 2 weeks, 6 weeks, and 3 months postoperatively.

### 2.5. Outcome Assessments

The outcomes were measured 2 weeks before the operation (T1), on the admission day (T2), on the discharge day (T3), at 2 weeks (T4), 6 weeks (T5), and 3 months (T6) follow-up postoperatively. The self-reported questionnaires were completed with the assistance of a study nurse and an orthopedic attending who were blinded to the study design. Clinical pain as the primary outcome measure was evaluated with the visual analog scale (VAS). The anxiety status was measured via the State-Trait Anxiety Inventory (STAI). The STAI is one of the most commonly used self-reporting measures for subjective anxiety, comprising two 20-item scales, assessing state (STAI-S) and trait (STAI-T) levels of anxiety, respectively. The functional outcomes were evaluated with American Knee Society scores (AKS) and the Western Ontario and McMaster Universities Arthritis Index (WOMAC).

### 2.6. Statistical Analysis

We performed an a priori calculation for sample size using G Power version 3.1. based on an analysis of variance (ANOVA) repeated measures analysis, between factors, of variance (group = 2, measurements = 4) with an effect size of 0.3. The sample size was calculated with a power of 80% (1-β error) at a significance level of 0.05. The calculating results revealed a minimum requirement of 54 patients in the study arms. We enrolled 36 patients in both groups because of an anticipated 20% case loss.

We also performed a post hoc power analysis as given α, sample size, and effect size; our results show that the study cases in this study can achieve the statistical test power > 80% by using repeated measures ANOVA, with factors.

The continuous variables were analyzed using the independent t-test, and the results were presented as a mean with standard deviation (SD). The categorical variables were analyzed using the Chi-square test, and the results were presented as numbers and percentages. A repeated measures generalized linear model analysis was used to evaluate series changes in VAS, STAI, AKS, and WOMAC. This model enabled each index parameter measurement from each participant as a separate observation and was adjusted for within-participants correlations. The interaction between the time and index parameters was shown in this generalized linear model. The multiple comparison adjustment of least squares means differences between groups by times using a Bonferroni test. A P value of <0.05 was considered to indicate a statistically significant difference. All statistical analyses were performed using SPSS (version 20.0 for Windows, SPSS Inc. Chicago, IL, USA).

## 3. Results

### 3.1. Patient Characteristics and Preoperative Assessment

The demographic characteristics and anthropometric data of both groups are illustrated in Table 1. A total of 72 people were eligible for the study, 36 in the intervention group and 36 in the control group. In total, 2 individuals, 1 in each group, were further excluded because of medical disease. There was no significant difference between the groups in age, gender, body mass index (BMI), or preoperative measures, including VAS, STAI-S, STAI-I, and AKS. The average WOMAC score in the intervention group was 55.1 ± 13.5, which is significantly higher than the control group (44.5 ± 11.1), indicating an inferior preoperative functional status in the intervention group. Further subgroup analysis revealed a significant difference in the WOMAC physical function subscale (*p* = 0.0363), but no significant difference in the pain and stiffness subscales.

### 3.2. IEP Reduced the Clinical Pain Intensity Pre-and Postoperatively

We calculated the intervention efficacy for patients either receiving IEP or usual rehabilitation programs and found that the intervention group exhibited a significant improvement in clinical pain, anxiety status, and knee function compared with the control group. Figure 2 shows the time course for a change of mean pain intensity from baseline to each time point. To evaluate the maintenance of the effect, we analyzed the mean pain intensity at each time point with a mixed model for repeated measurements. The difference in the least square means between the intervention group and control group was significant on the admission day (T2), the discharge day (T3), at 2 weeks (T4), and after 6 weeks (T5) postoperative follow-up.

### 3.3. IEP Reduced Both State and Trait Anxiety in TKR

We use STAI to assess the anxiety symptoms in patients who underwent TKR. The time-course changes in state and trait anxiety scores in both groups are shown in Figure 3a,b. We compared the changes of STAI-S at different times and found that the difference in the least square means between the intervention group and control group was significant at T2, T3, T4 and T5. Although there was significantly less reduction in STAI-S from T1 to T6 in the intervention group, the absolute score in both groups was not significantly different (20.14 in the intervention group vs. 20 in the control group). The state anxiety was significantly reduced by an IEP at admission. The average STAI-S in the intervention group significantly decreased from 33.9 at T1 to 29.21 at T2 (*p* < 0.001) but significantly increased from 29.6 at T1 to 31.7 at T2 in the control group (*p* < 0.001). Moreover, the STAI-S remained higher than baseline in the control group on discharge day (*p* = 0.024 from T1 to T3). The analysis of STAI-I at each time point also indicated that the difference in the least square means between the intervention group and control group was significant at each time point (*p* < 0.001 at T2, T3, T4 and T5, *p* = 0.006 at T6).

### 3.4. IEP Enhanced the Physical Function after TKR

We used AKS and WOMAC to assess the physical functions after TKA in the two groups. We calculated and analyzed the differences in AKS changes between the two groups, showing that the difference in the least square means between the intervention group and control group all reached significant differences (*p* = 0.001 at T3 and *p* < 0.001 at T4, T5 and T6) at each time point (Figure 4). Similar results were also obtained in WOMAC scores (Figure 5a). The differences in the least square mean between the two groups were all significant (*p* = 0.001 at T3 and *p* < 0.001 at T4, T5 and T6). The average WOMAC score was significantly reduced at discharge in the intervention group, whereas there was no significant difference in the control group. Although there was a significantly higher WOMAC score in the intervention group at baseline, the absolute WOMAC score of the intervention group was significantly lower than the control group at T3, T4, and T5 (Figure 5b).

## 4. Discussion

Pain and function at 3 months post-TKA are strongly associated with long-term clinical outcomes and patient satisfaction [9,10]. Active postoperative rehabilitation exercises in the acute phase can help restore muscle strength and increase the range of motion, resulting in better functional recovery. Preoperative patient education has been identified as an important part of joint arthroplasty to decrease postoperative pain, reduce anxiety, and decrease the length of hospital stay. Motivational-based care navigation can promote behavior change and may help increase adherence to rehabilitation programs after discharge. This study supports the hypothesis that an IEP consisting of preoperative education, prehabilitation, perioperative group education, and a care navigation program helps patients achieve better pain control, a lower anxiety status, and earlier functional recovery.

Pain is a multidimensional experience, and the response to postoperative pain varies in each individual. Preoperative patient education has been considered an effective adjuvant regimen to decrease postoperative pain in joint arthroplasty. Moreover, increasing the patient’s understanding of pain neuroscience and pain management before surgery can help obtain better pain relief postoperatively [11,12]. Most patients undergoing TKA are elderly and have a different level of education, so preoperative education with standard manuals may not be fully understood by patients. Even in highly educated countries, 77% of individuals reportedly had unfulfilled knowledge expectations [6]. Another study found that more than 30% of TKA patients had unfulfilled expectations of functional outcomes after 1 year [13]. We considered that a patient-specific modification of preoperative education with a joint-specific education might be beneficial for understanding the treatment process and managing expectations. In this study, we conducted a one-on-one preoperative education and prehabilitation exercise course by an experienced nurse and a physical therapist for each individual in the intervention group. As a result of the intervention, patients experienced less pain intensity before discharge compared with the control group. We could not determine if this result was due to preoperative education or to the combination of education and prehabilitaion, but the personalized preoperative education and prehabilitation exercise did appear to be beneficial for early pain relief after surgery.

Preoperative psychological distress is common in patients undergoing joint replacement surgery; it has been reported that approximately 25% of patients undergoing hip or knee arthroplasty have preoperative psychological distress [14]. It is well documented that psychological distress negatively influences postoperative physical function and pain [14,15,16,17,18,19]. A systematic review by Alattas et al. recently reviewed the functional outcomes of TKA-related preoperative psychological factors and demonstrated that symptoms of depression or anxiety were risk factors predicting a poorer outcome after TKA [19]. Patients with psychiatric disorders were associated with higher hospital costs and postoperative pain-related symptoms [17]. Furthermore, disease-specific anxiety at discharge was a significant risk factor for inferior outcomes at 6 months postoperatively [18]. Consistent with the previous meta-analysis, which indicated that preoperative education moderately decreased preoperative anxiety [20], our results also showed that IEP intervention positively deceased preoperative state anxiety at admission, whereas the anxiety level was significantly increased in the control group after the usual preoperative education, indicating better efficacy in attenuating anxiety with one-on-one preoperative patient education and prehabilitation compared with standard treatment. Although several studies indicated that preoperative education reduced preoperative anxiety, the association between postoperative anxiety and preoperative patient education remains unclear. From our results, we noted that the anxiety score declined gradually after TKA, but the average anxiety score of the control group remained higher than the preoperative level at discharge. Conversely, the anxiety level in the intervention group was markedly reduced after TKA, presenting significantly lower anxiety scores compared with the baseline level at discharge. A recent meta-analysis reported that there was no significant difference in postoperative anxiety reduction after preoperative education [21]. We consider that the possible reasons for this difference include the group education class before discharge, reinforcing the postoperative rehabilitation exercises, and allowing patients to compare and share their experiences with peers, which may help to decrease postoperative anxiety.

In our study, we also found that both patient- (WOMAC) and physician-reported (AKS) functional scores significantly improved after the IEP. Prehabilitation is thought to increase functional capacity before TKA and be of benefit in reducing hospital stay and improving functional recovery [21,22]. A recent meta-analysis comparing the clinical outcomes in terms of pain, function, muscular strength, anxiety, and length of hospital stay in patients undergoing hip or knee arthroplasty after preoperative intervention found that prohabilitaion positively influenced outcomes [21]. However, two systematic reviews demonstrated inconsistent results and concluded that prehabilitation has no significant postoperative benefits in functional recovery [23,24]. We considered that the improvement in functional outcome is not solely based on prehabilitation. Because most patients are elders, they may not fully understand or remember all the rehabilitation exercises. Besides prehabilitation, the IEP also included an in-hospital group education class and a postoperative care navigation program. Multi-model education programs can enhance memory and encourage continuous exercise to facilitate functional recovery.

There are limitations to this study. First, this study was conducted in two institutions in the same medical system. Although the surgical procedures, clinical pathway, and postoperative rehabilitation were the same, the results may still be affected by differences in medical staff and hospitalization environment. Second, there were significantly higher WOMAC scores in the intervention group which may cause an interpretation bias. Patients with higher preoperative WOMAC scores may gain greater decreases in WOMAC scores following TKA [25,26]. However, this does not mean that the patients with the worst WOMAC will have better absolute outcomes. A poor preoperative function is well reported to be associated with poor outcomes in TKA [27,28]. Despite a marked improvement in patients with higher preoperative WOMAC scores, the absolute physical function score remained lower than for those with higher preoperative physical function [26]. In our study, there was not only a marked improvement between the preoperative and postoperative WOMAC scores, but the absolute WOMAC scores were also significantly higher in the control group than in the intervention group. This result indicates a positive impact of IEP on functional recovery. Third, the study was designed to assess the efficacy of multimodal education programs on the outcomes of TKA. We could not distinguish the exact clinical efficacy of each element in the education program.

## 5. Conclusions

We found that an IEP combining preoperative education and prehabilitation, an in-hospital group education class before discharge, and postoperative care navigation was effective in reducing postoperative pain, decreasing preoperative and postoperative anxiety, and facilitating functional recovery in the first 3 months following TKA. Although this study focused on individuals undergoing TKA, we believe that a similar effect can be achieved in patients undergoing total hip replacement. However, the cost-effectiveness of this program was not assessed in the current study. Further studies are needed to determine the economic benefits of this education program.

## Figures and Tables

**Figure 1 jpm-12-00719-f001:**
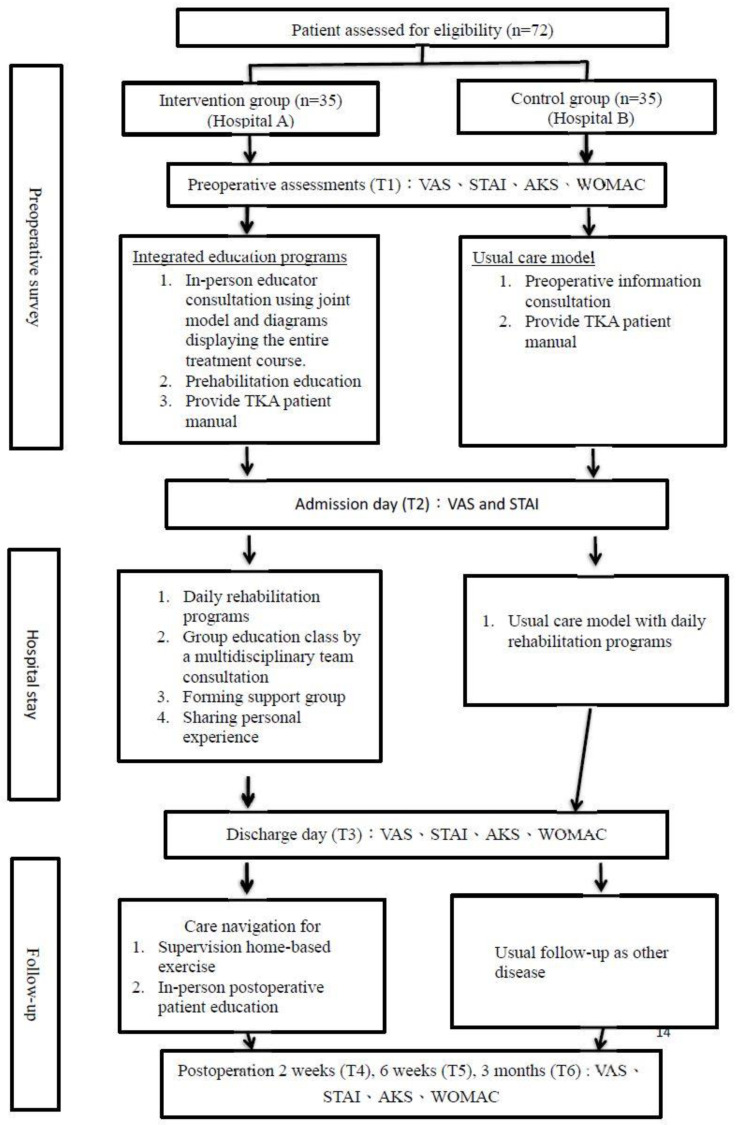
Detailed flowchart of the IEP (integrated education program).

**Figure 2 jpm-12-00719-f002:**
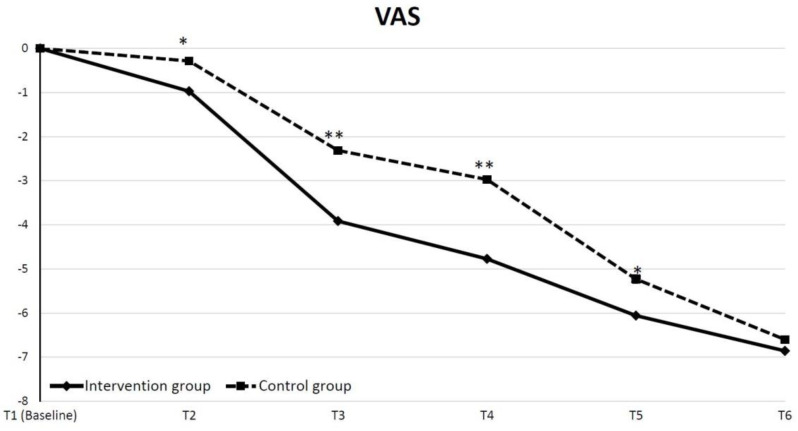
Time course of mean change in pain intensity from baseline to each checked time point (* *p* < 0.05, ** *p* < 0.01.)

**Figure 3 jpm-12-00719-f003:**
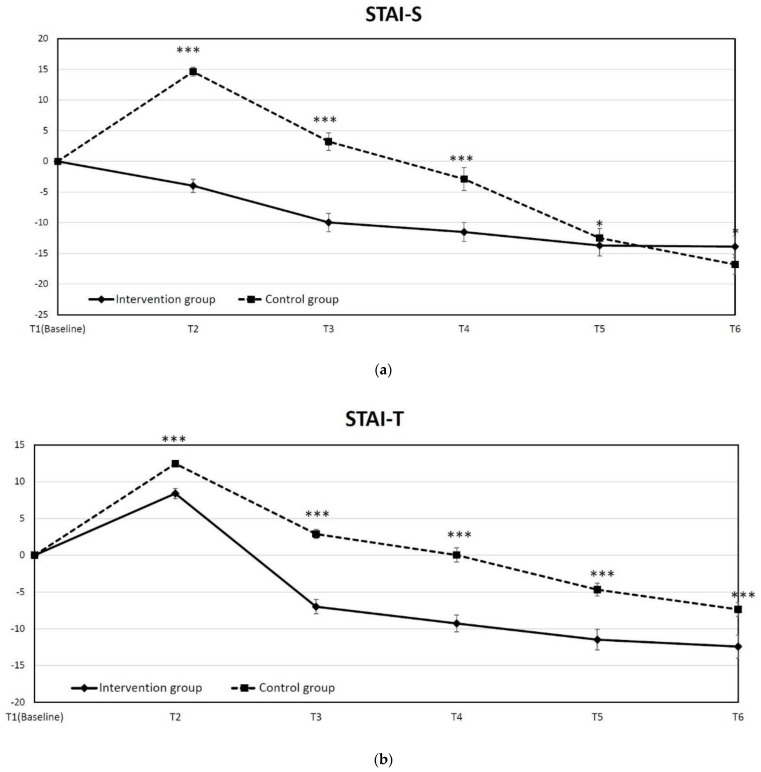
Time course of mean change in anxiety state from baseline to each checked time point: (**a**) state anxiety (State-Trait Anxiety Inventory, STAI-S), (**b**) trait anxiety (State-Trait Anxiety Inventory, STAI-T). (* *p* < 0.05, *** *p* < 0.001).

**Figure 4 jpm-12-00719-f004:**
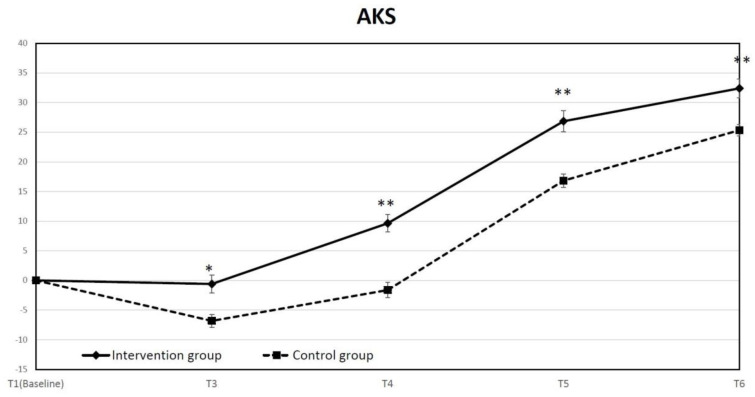
Time course of mean change in American Knee Society Score (AKS) from baseline to each checked time point. (* *p* < 0.05, ** *p* < 0.01).

**Figure 5 jpm-12-00719-f005:**
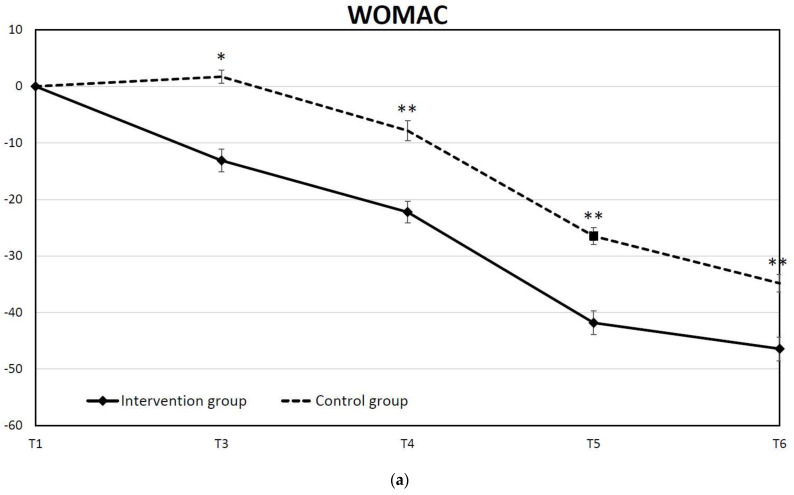
(**a**) Time-course of mean change in WOMAC from baseline to each time point. (**b**) The absolute WOMAC score at each time point in both groups. (* *p* < 0.05, ** *p* < 0.01; *** *p* < 0.001).

**Table 1 jpm-12-00719-t001:** The demographic data in participants.

	Intervention Group (*n* = 35)			Control Group (*n* = 35)			
Variables	Mean	SD	Rang	Mean	SD	Rang	*p*
Age	73.5	5.3	68–85	74.4	5.4	65–86	0.521
Gender							0.073
Male	10	28.6%		4	11.4%		
Female	25	71.4%		31	88.6%		
BMI	28.0	3.2	23.2–38.1	26.6	4.0	17.8–40.1	0.058
Education level							0.068
Illiterate	7	20.0%		11	31.4%		
Elementary school	13	37.1%		17	48.6%		
Junior high school	5	14.3%		2	5.7%		
Senior high school	9	25.7%		4	11.4%		
College or above	1	2.9%		1	2.8%		
Pain (VAS)	7.1	1.8	4–10	7.3	1.4	5–10	0.585
State anxiety	33.9	10.6	20–54	29.6	5.5	20.40	0.178
Trait anxiety	33.9	10.5	20.57	30.0	6.8	21–50	0.208
AKS	56.6	9.5	28–73	59.2	7.3	38–70	0.202
WOMAC	51.1	13.5	24–88	44.5	11.1	23–88	0.009 *

* *p* < 0.05.

## Data Availability

The datasets generated and analyzed during the current study are available from the corresponding author on reasonable request.

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
