# Peer review of "The Effects of a Patient-Specific Integrated Education Program on Pain, Perioperative Anxiety, and Functional Recovery following Total Knee Replacement"

_jpm, 2022, doi:10.3390/jpm12050719_

Round 1

Reviewer 1 Report

Title: The effects of a patient‐specific integrated education program on pain, perioperative anxiety, and functional recovery following total knee replacement.

This article seems well built and brings evidence of a physiological phenomenon not yet fully understood and that certainly deserves further study.

this paper cannot be evaluated ....

Anthropometric data are missing (please to include this information’s)

Informed consent and ethical Committee is missing

ANOVA's results are missing

Reviewer 2 Report

Line 30/ 31: Please include the p-value.

Line 55-62: was the IEP performed as an outpatient procedure or inpatient?

Line 65: what is a quasi-experimental research design?

Line 72: where are patients operated by the same surgeon?

Please reference the flow chart in the methods section.

Seems you discharge the patients following the IEP 2 days later than the control group.

Line 132-134: Your power analysis revealed a minimum of 54 patients however, you only enrolled 36 patients? Please explain. I would suggest to perform a post hoc power analysis. 

Figure 1/ Line 303: you mentioned that the discharge day was differently between the two groups. Please adjust. 

Where does the difference in average WOMAC scores come from: (in the intervention group was 55.1±13.5, which is significantly higher than the control group (44.5±11.1))

Line 293-294: Other publications show that poor preoperative outcome will improve significantly postoperative outcome.

The higher the preoperative WOMAC scores the better the improvement postoperatively. 

What is the reason? I assume the later discharge day is an important criteria.

Please show the values in a table. 

Have you performed a subgroup analysis including strength and range of motion?

Please include the different demographics and show if there are any differences?

Line 312-314: It is not relevant that you did not assessed the cost effectiveness. This was not part of your aim/ hypothesize Please delete this sentence.

Reviewer 3 Report

Please check the following and correct it.
  1. Problems with grammar and expression. please revise it. 
    2. Also, Please revise the paper according to the journal form.
    3. Effect size 0.3 is small. What do you think?
    4. Please fill out the reliability and validity of the all evaluation tool.
    5. Table 1 cannot be found. Please check it out.
    6. Figure 1 targets 72 people, with 35 people assigned per group. Please write down the reason.

Round 2

Reviewer 1 Report

The authors improved the mani document about the ethical committee and informed consent,  after first revision. Anyway the statistical analysis is very poor: the ANOVA (2(groups) x 5 (times point, as showed in the figures)) and Effect size with sample power is needed to be accepted.

Author Response

Reviewer 1

Thank you for giving me the opportunity to submit a revised draft of my manuscript titled ”The effects of a patient-specific integrated education program on pain, perioperative anxiety, and functional recovery following total knee replacement” to “Journal of Personalized Medicine”. We appreciate the time and effort that you and the reviewers have dedicated to providing your valuable feedback on our manuscript. We are grateful to the reviewers for their insightful comments on our paper. We have been able to incorporate changes to reflect most of the suggestions provided by the reviewers. We have highlighted the changes within the manuscript.

Comments and Suggestions for Authors

The authors improved the mani document about the ethical committee and informed consent, after first revision. Anyway the statistical analysis is very poor: the ANOVA (2(groups) x 5 (times point, as showed in the figures)) and Effect size with sample power is needed to be accepted.

Author response and actions:

Thank you for pointing this out. The power analysis revealed a minimum of 54 patients, which means 27 patients in each group. We included a total of 70 patients in this study and our current study design met the power requirement in our analysis. We also performed a post hoc power analysis. Our results show that the study cases in this study can achieve the statistical test power (80%) that we set before the experiment. Our analysis showed that the effect size 0.3 is appropriate for our study aim.

Our results show that the study cases in this study can achieve the statistical test power> 80% by using G Power version 3.1 post hoc power test (ANOVA: repeated measures, between factors) that we set after the experiment.

VAS

Intervention Group

Control Group

Difference

β

SE

β

SE

p-value by t-test

Effect size f (repeated measures)

Power (1-β)

0.00

0

0.00

0

 T2-T1

-0.9710

0.23

-0.2860

0.16

0.0160

0.2954

0.8034

 T3-T1

-3.9140

0.31

-2.3140

0.32

0.0006

0.4300

0.9914

 T4-T1

-4.7710

0.33

-2.9710

0.31

0.0002

0.4733

0.9985

 T5-T1

-6.0570

0.29

-5.2290

0.23

0.0296

0.2656

0.8073

 T6-T1

-6.8570

0.31

-6.6000

0.22

0.5057

0.0800

0.1388

We thank the reviewer for pointing this out. We didn’t fully understand the means of reviewer’s comment. However, we have tried our best to rewrite the “2.6. Statistical analysis” to make it clearer.

2.6. Statistical analysis

We performed an a priori calculation for sample size using G Power version 3.1. based on an analysis of variance (ANOVA): repeated measures analysis, between factors, of variance (group=2, measurements=4) with an effect size of 0.3. The sample size was calculated with a power of 80% (1 – β error) at a significance level of 0.05. The calculating results revealed a minimum requirement of 54 patients in the study arms. We enrolled 36 patients in both groups because of an anticipated 20% case loss.

We also performed a post hoc power analysis, as given α, sample size, and effect size; our results show that the study cases in this study can achieve the statistical test power >80% by using repeated measures ANOVA, with factors.

The continuous variables were analyzed using the independent t-test and the results were presented as mean with standard deviation (SD). The categorical variables were analyzed using Chi-square test and the results were presented as numbers and percentage. A repeated-measures generalized linear model analysis was used to evaluate series changes in VAS, STAI, AKS and WOMAC. This model enabled each index parameter measurement from each participant as a separate observation and was adjusted for within-participants correlations. The interaction between the time and index parameters were shown in this generalized linear model. The multiple-comparison adjustment of least squares means differences between groups by times using a Bonferroni test. A P value of < 0.05 was considered to indicate a statistically significant difference. All statistical analyses were performed using SPSS (version 20.0 for Windows, SPSS Inc. Chicago, USA).

The revised manuscript: Page 4 ; Line 134-154

2.6. Statistical analysis

We performed an a priori calculation for sample size using G Power version 3.1. based on an analysis of variance (ANOVA): repeated measures analysis, between factors, of variance (group=2, measurements=4) with an effect size of 0.3. The sample size was calculated with a power of 80% (1 – β error) at a significance level of 0.05. The calculating results revealed a minimum requirement of 54 patients in the study arms. We enrolled 36 patients in both groups because of an anticipated 20% case loss.

We also performed a post hoc power analysis, as given α, sample size, and effect size; our results show that the study cases in this study can achieve the statistical test power >80% by using repeated measures ANOVA, with factors.

The continuous variables were analyzed using the independent t-test and the results were presented as mean with standard deviation (SD). The categorical variables were analyzed using Chi-square test and the results were presented as numbers and percentage. A repeated-measures generalized linear model analysis was used to evaluate series changes in VAS, STAI, AKS and WOMAC. This model enabled each index parameter measurement from each participant as a separate observation and was adjusted for within-participants correlations. The interaction between the time and index parameters were shown in this generalized linear model. The multiple-comparison adjustment of least squares means differences between groups by times using a Bonferroni test. A P value of < 0.05 was considered to indicate a statistically significant difference. All statistical analyses were performed using SPSS (version 20.0 for Windows, SPSS Inc. Chicago, USA).
